# FSTS: A FEATURE SPACE TRANSFER SELECTION METHOD FOR DATA

## ABSTRACT

The performance of deep learning image models is often largely determined by data quality. However, high-quality data is often scarce and difficult to collect. The data exchange platform provides a promising solution for obtaining image training samples, but the actual data exchange must consider the costs associated with data collection, storage, and training. This article proposes a Feature Space Transfer Selection (FSTS) method for identifying core data subsets that are crucial for model training. The proposed method extracts feature vectors from the source and target datasets, calculates class centroids from the reference (source) dataset, and ranks the target samples based on their similarity to these class centroids. Then select the core dataset from the target data based on the similarity ranking. The experimental results show that FSTS outperforms prior state-of-the-art approaches and effectively helps users select the core set of training data, which helps improve the efficiency of model training and overall performance.

## 1 INSTRUCTION

In deep learning image model training, the quality and diversity of the training dataset directly affect the effectiveness and generalization ability of the model training. However, high-quality data often accounts for a relatively small proportion of the entire dataset, and the production of high-quality datasets requires a significant amount of cost. In practical applications, collecting a sufficient number of high-quality training samples is a challenging task. Factors such as lighting conditions, shooting angles, and environmental noise during image capture may affect data quality, resulting in blurry, noisy, or obstructed images. Data exchange has become a promising solution to the problem of obtaining high-quality image data, providing a shared space and mechanism that allows different data holders to exchange data resources. These features have made significant contributions to enriching the training dataset and improving model performance. However, considering the costs of transmission,storage, and data exchange themselves, users must consider the following key cost factors when purchasing data through data exchange platforms:

- Data acquisition cost: In the field of autonomous driving, for instance, the cost of acquiring a single image sample for training a classification model is approximately USD $0.70. Acquisition cost is a critical consideration in constructing large-scale datasets, as it directly affects the overall R&D budget.

- Data storage cost: A typical video dataset for autonomous driving training, such as BDD100K (Yu et al., 2018), can reach a size of 1.8TB. The storage of large-scale datasets necessitates considerable hardware resources, making storage costs a pivotal concern in data management.

- Model training computational cost: For example, training an autonomous driving image classification model using EfficientNet-B7 (Tan & Le, 2019) on the ApolloScape (Huang et al., 2018) dataset takes approximately 20 hours on an A100 GPU. This process consumes considerable hardware resources, electrical power, and time, all of which need to be accounted for during project budgeting.

Therefore, data exchange platforms should include efficient data filtering mechanisms, qualitative and quantitative screening, storage, and transmission of data to help users ensure data quality and

improve model performance, while more effectively controlling training costs. This problem can be divided into two parts to solve:

1. Calculate the sample quality distribution in a batch of data samples and identify the location of the best quality data sample.

2. Divide the highest quality data samples into intervals and filter the data to form the smallest dataset available for model training.

In this study, we propose a new method called Feature Space Transfer Selection (FSTS), which selects the core dataset based on feature extraction and similarity calculation. This method incorporates the feature representations of candidate samples and the positions of reference samples in the feature space of the candidate set into the reference, thereby avoiding the problem of containing irrelevant samples that rely solely on sample similarity or distance. The selection process does not require model training or gradient or error vector calculations, and only a small number of high-quality samples are needed to significantly improve model performance. The main contributions of this paper are as follows.

1. Proposed the FSTS method - an effective data selection method designed for data exchange platforms, which helps models train efficiently with small sample sizes, and introduced the design and implementation principles of the FSTS module.

2. We compared the performance differences between FSTS and other data selection methods trained on the core dataset on the open-source KITTI Road (Geiger et al., 2012) and CIFAR-10 datasets (Krizhevsky, 2009). Introducing artificial noise into CIFAR-10 images and labels to evaluate the robustness of FSTS under various noise conditions.

## 2 RELATED WORKS

Image Quality Assessment based data selection (IQA) refers to the process of quantifying image quality using various algorithms and metrics to evaluate its similarity to a reference image or whether it meets human perceptual standards. Image quality analysis technology has been widely applied in many fields, including image enhancement and retrieval. The traditional IQA method is mainly based on low-level visual features. Common methods include Structural Similarity Index (SSIM) and Peak Signal to Noise Ratio (PSNR) (Horé & Ziou, 2010), as well as Multi Scale SSIM (MS-SSIM) (Richter & Kim, 2009).

In recent years, deep learning has significantly advanced the development of IQA field.Models based on Convolutional Neural Networks (CNN) have demonstrated excellent accuracy by directly learning high-level image features from data(Xie et al., 2022). IQA methods, similar in principle to CNN, can conduct calculations starting from the high-level features of image data(Alshowaish et al., 2022; Wang et al., 2023).

Both traditional IQA methods and deep learning-based IQA methods have their own advantages and limitations. Traditional methods such as SSIM and PSNR are intuitive and easy to understand, with high computational efficiency. They are suitable for initial quality inspection and perform particularly well in resource-constrained environments. However, due to their reliance on low-level features, these methods often fail to capture deeper semantic information and contextual information, which in turn leads to inaccurate evaluation results. In contrast, deep learning-based IQA models can extract richer and more advanced feature representations, and exhibit better performance in more complex scenarios. Nevertheless, deep learning-based IQA models usually have high computational complexity and consume substantial computational resources. Therefore, a comprehensive approach is needed to reduce computational resources while maintaining its performance. Core dataset selection refers to the process of selecting high-quality, discriminative small-sample datasets from massive amounts of data for model training. This process can not only reduce the consumption of computational resources but also lower the storage and transmission costs of data exchange platforms. A study (Toneva et al., 2018) proposed a "forgetting" algorithm based on the observation that the model is prone to forgetting some samples during the training process. By eliminating frequently forgotten data points, this method preserves the most advantageous samples for the model. Mirzasoleiman et al. (2019) proposed a gradient matching based core set selection method called Craig (CoResets for Accelerating Incremental Gradient descent). The core idea is to use greedy algorithms

to select a weighted subset from the entire dataset, so that the weighted gradient of this subset is close to the average gradient of the entire dataset. This method ensures that models trained on the core dataset achieve performance comparable to models trained on the entire dataset. The proxy model method (Coleman et al., 2019) provides an alternative approach to data selection. Build simplified models by reducing the number of layers and hiding units or using basic architectures. Then train the model on the target dataset to approximate the data distribution and extract significant features, using data selection strategies such as maximum entropy uncertainty sampling, greedy K-centers, and forgetting events to identify representative subsets. The proxy model method significantly reduces selection time and achieves rapid iteration in the model development process. Killamsetty et al. (2020) proposed Glister (a GeneraLIzation based data Subset selecTion for Efficient and Robust learning framework ), a noise aware kernel set selection method based on a two-layer optimization framework that can automatically identify high-value samples that are most effective in improving model performance, and use validation feedback to detect and remove noisy samples.

These methods have their own advantages, but they still face the following challenges:

- Scenario dependency: Many methods are scenario specific, and their performance will significantly decrease when migrating to new scenarios.

- Poor stability: These methods often lack robustness under constantly changing requirements or new use cases. Adjusting them to adapt to new tasks may be difficult, making them less suitable for actual deployment.

- High computational cost: Quantifying the contribution of each sample to the training process typically requires intensive computation.

## 3 FSTS METHODOLOGY

### 3.1 SYSTEM OVERVIEW

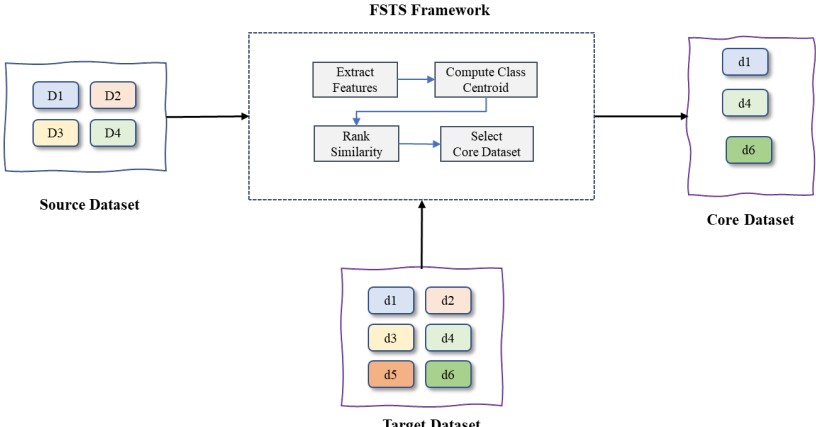

Figure 1: System Overview.

To address the above issues, this article proposes a new data selection method (FSTS). As shown in Fig.1, the FSTS method enhances the sample selection process in the data exchange platform through innovative strategies inspired by the transfer learning theory framework(Li et al., 2020). Transfer learning is the process of learning features and patterns from a source dataset and applying them to a new target dataset. The large-scale dataset used to train a model is called the source dataset, which comes from different but related tasks or domains. In traditional transfer learning, the source dataset is selected based on the features of the target dataset. The logic of the FSTS method is the opposite, selecting the target dataset from the source dataset to construct the data required for model training.

The FSTS method operate steps are as follows:

1. Extract Source Feature: A deep learning model is employed to extract features for each class in the source dataset.

2. Compute Feature Centroid: For each class, the centroid of the feature vectors is computed to obtain a representative feature for that class.

3. Extract Target Feature: The same deep model is used to extract features from samples in the target dataset.

4. Compute Similarity: The similarity between the extracted target features and the corresponding source feature centroids is calculated.

5. Rank: Target samples are ranked according to their similarity scores with respect to the source class centroids.

6. Select Coreset: Based on the similarity ranking and a predefined selection policy, samples are chosen to construct a representative core dataset.

FSTS selects samples highly correlated with the source dataset from the target dataset through transmission in the feature space. This method not only considers the intrinsic feature representation of the samples, but also their relative positions in the feature space with respect to the source samples. This can reduce the risk of selecting irrelevant samples when relying solely on pairwise similarity or distance measures. Even under domain shift conditions, FSTS prioritizes samples that are semantically aligned with the source dataset through feature space similarity, thereby maintaining robustness. In addition, due to the fact that the FSTS method only relies on feature extraction and similarity calculation, without the need for model training or calculating gradients or error vectors during the training process, its computational complexity is lower.

## 3.2 IMPLEMENTATION

**Dataset.** FSTS involves three types of datasets: the source dataset, the target dataset, and the core dataset. To suit the context of data exchange platforms, we redefine the source and target datasets under the transfer learning framework (Zhuang et al., 2019). The source dataset comprises validated and curated training samples that are directly usable for deep learning. These samples consist of image data and their corresponding labels. Although the source dataset is of high quality, it is typically limited in size. The target dataset refers to datasets available for selection on data exchange platforms, often used in domain-specific model training. For example, the Guizhou Big Data Exchange in China provides curated datasets for domains such as healthcare, meteorology, and transportation(Guiyang Big Data Administration, 2025). These datasets are either collected through devices or generated synthetically using computational methods and algorithms (Nikolenko, 2019), with annotations provided, making them suitable for training, testing, and validating industry-specific deep learning models. The core dataset is a smaller, representative subset selected from the target dataset (Xia et al., 2023b) , designed to have features statistically aligned with the source dataset. This subset can be combined with the source dataset to create an enriched training dataset that enhances model performance. The user selects relevant data from the target dataset via the platform to form this core dataset.

Let the source dataset be denoted as $\mathbb{S} = \{S_1, S_2, \cdots, S_m\}$, where each $S_j$ represents a category and $m$ is the number of categories. The target dataset is $\mathbb{D} = \{D_1, D_2, \cdots, D_n\}$, similarly defined. The core dataset is denoted as $\mathbb{D}^* = \{\hat{D}_1, \hat{D}_2, \cdots, \hat{D}_l\}$, where $\hat{D}_j \in D_j \in \mathbb{D}$, $l \leq n$ and $l \leq m$.

**Features.** Features (Lu et al., 2023) are informative patterns extracted from raw data that help models interpret and solve problems. Features are critical for enabling models to learn data patterns and make predictions. They are broadly categorized into low-level features and high-level features.

Low-level features (Nixon & Aguado, 2020) pertain to raw physical characteristics, such as pixel values in images or amplitude in audio signals. In contrast, high-level features (Liu et al., 2019) are abstract representations formed progressively through deep network layers, such as object shapes or semantic meaning in text. In this paper, we employ high-level features to select sample data.

In deep learning, high-level feature extraction is mainly accomplished using a deep model, e.g., ResNet-50 or VGG-16. Let the deep model $f(x; \theta)$ be decomposed into a feature extraction function $\phi(x; \theta')$ and an output function $h(\cdot; \theta'')$, and can be expressed as $f(x; \theta) = h(\phi(x; \theta'); \theta'')$. The

output of the penultimate layer (i.e., the feature layer) with input $x$ denoted as:

$$z = \phi(x; \theta') \tag{1}$$

Here, $\theta' \subseteq \theta$ denotes the subset of the model parameters excluding the final output layer.

Feature extraction is performed via $\phi(x; \theta')$. The features extracted from the source dataset $\mathbb{S} = \{S_1, S_2, \cdots, S_m\}$ are $\mathbb{S}^z = \{z_1 = \phi(S_1; \theta'), \cdots, z_m = \phi(S_m; \theta')\}$. Similarly, for the target dataset $\mathbb{D} = \{D_1, D_2, \cdots, D_n\}$, the extracted features are $\mathbb{D}^z = \{z_1 = \phi(D_1; \theta'), \cdots, z_n = \phi(D_n; \theta')\}$.

**Compute the Class Centroid.** A class centroid, computed from all feature vectors of that class, is a representative point in feature space for a specific class. Common methods for centroid computation include mean and median (Sun & Qu, 2014).

- Mean: The arithmetic average of all vectors, assigning equal weight to each. Suitable for normally distributed data, but sensitive to outliers.
- Median: The middle value in the sorted list of vectors. More robust to outliers, though computationally more intensive due to sorting.

Considering the variability in high-level features, such as those in autonomous driving datasets, we adopt the median method to ensure robustness. The centroid for class $j$ in the source dataset is computed as:

$$c_j^{(med)} = \text{median}(z_j), \quad j = 1, 2, \cdots, m \tag{2}$$

**Compute Similarity.** Data similarity measures the correlation between a target sample and a class centroid, typically via distance metrics (Charikar, 2002; Kanungo, 2024; Sorscher et al., 2022a; Xia et al., 2023a). Common methods include:

- Euclidean Distance: Straight-line distance between two points. Effective for low-level data but suffers from the "curse of dimensionality" (Ng, 2002) in high-level settings.
- Cosine Similarity: Measures the angle between two vectors. Values near 1 imply high similarity. Cosine similarity is scale-invariant and well-suited for high-level data, thus widely used for text and image vectors (Srikaewsiew et al., 2022).

Given image size differences between source and target samples, we use cosine similarity to compute distances from each target sample to the corresponding class centroid:

$$Cosine\left(z_j, c_j^{(med)}\right) = \left\{ \frac{d_{i,j} \cdot c_j^{(med)}}{\|d_{i,j}\| \|c_j^{(med)}\|} \right\}_{i=1}^{\|z_j\|} \tag{3}$$

where $d_{i,j} \in z_j \in \mathbb{D}^z$ is a sample feature vector from class $j$ in the target dataset.

**Select Coreset.** Once similarities are calculated, each class's samples are sorted in descending order based on cosine similarity to form the ranked list. A data pruning method proposed by Sorscher et al. (2022b) enables efficient sample selection through self-supervised learning. The author proposes that for the self-supervised prototype metric (k-means cosine distance), when the number of clusters $k$ varies between $200$–$10000$ ($\approx 0.2\times$–$10\times$ original categories), the Top-5 accuracy remains at $\approx 90\%$, significantly better than extreme values ($k \leq 10$ or $k \geq 50k$). By retaining $40\%$–$60\%$ of samples with "moderate aggregation degree" (i.e. neither extreme redundancy nor extreme outliers), the performance of the entire dataset can be approximated. The approach first uses the SWAV model with a ResNet-50 backbone to extract image features, then applies $k$-means clustering within the feature space for each class. The cosine distance between each sample and its nearest class prototype is computed as a difficulty metric. By dynamically selecting samples whose scores fall within the 40% to 60% of the global distribution, the method achieves 90.27% accuracy on ImageNet using only 60% of the data—compared to 90.85% when using the full dataset. Building on this, we propose a generalized coreset selection strategy. Assuming a selection ratio $R$, the number of selected samples in each class is $k = \lfloor round(\|z_j\| \times R) \rfloor$. The selection range is determined by:

$$a = round\left(\frac{\|z_j\| - k}{2}\right), \quad low = a, \quad upper = \|z_j\| - a \tag{4}$$

Thus, the selected core dataset for each class is:

$$\hat{\mathbb{D}} = \{D_{low}, \cdots, D_{upper}\} \tag{5}$$

In Appendix A.1, we further investigate three selection strategies—Closest, Farthest and Two-ends, all of them measure similarity to the class centroid.

# 4 EVALUATION

## 4.1 EXPERIMENTAL SETUP

**Dataset.** We evaluate the proposed FSTS method using KITTI Road and CIFAR-10 datasets.

The KITTI Road dataset, widely used for autonomous driving research, provides 289 annotated images for road/lane detection and semantic segmentation tasks. We construct the source dataset by selecting 29 images covering key urban scenarios: unmarked urban roads (uu), urban roads with single marked lanes (um), and urban roads multiple marked lanes (umm). The remaining 260 images form the target dataset.

The CIFAR-10 dataset is a widely used benchmark for general-purpose image classification in machine learning and computer vision. It consists of 10 classes, with 50,000 training images and 10,000 test images in total. To construct the source dataset, we select 1,000 images per class from the training set, ensuring high clarity, optimal viewing angles, and no occlusion. All selected images comprehensively represent the structural characteristics of the corresponding objects. The remaining 4,000 images are used as the target dataset.

**Baselines.** To assess the effectiveness of FSTS, multiple data selection methods act as baseline for comparison. We use (1) Random; (2) Craig; (3) Glister; (4) Graph cut; (5) Cal; (6) Forgetting. Due to the limited pages, we provide (7) Closest;(8) Farthest;(9) Two-ends in section A.1 and A.2 in Appendix.

**Implementation Details.** Core datasets containing 10%, 20%, 30%, 40%, and 50% of the target dataset are selected using FSTS and the baseline methods. For the KITTI Road dataset, UNet is used as the training model, and two standard segmentation metrics—Intersection over Union (IoU) and Dice Score (Dice)—are adopted for performance evaluation (Ngoc et al., 2023). For CIFAR-10, VGG-16 is used as the training model, and test accuracy is employed as the evaluation metric. Data selection is implemented using Python 3.8.10. Model training is conducted on an NVIDIA GeForce RTX 2080 Ti GPU using PyTorch (Paszke et al., 2019). Each training job uses a batch size of 256 and an SGD optimizer with momentum 0.9, weight decay 5e-4, and an initial learning rate of 0.1, which is reduced by a factor of 5 at epochs 60, 120, and 160. Each experiment is repeated 5 times with different random seeds, and the average IoU, Dice and test accuracy are reported.

## 4.2 EXPERIMENTAL RESULTS AND ANALYSIS

Table 1: IoU on KITTI Road

| Method/Ratio | 10% | 20% | 30% | 40% | 50% |
|---|---|---|---|---|---|
| Random | 85.71±0.55 | 89.55±0.03 | 90.58±0.39 | 92.60±0.19 | **94.50±0.12** |
| Craig | 85.92±0.98 | 89.56±0.37 | **91.55±0.11** | 92.09±0.20 | 93.85±0.31 |
| Glister | 85.85±0.79 | 88.20±0.43 | 90.33±0.41 | 91.69±0.21 | 93.20±0.57 |
| Graph Cut | 86.69±0.31 | 87.33±0.45 | 90.77±0.03 | 91.76±0.44 | 93.04±0.31 |
| Cal | 84.23±0.56 | 88.78±0.10 | 90.86±0.23 | 92.68±0.09 | 93.90±0.53 |
| Forgetting | 86.25±0.47 | 89.49±0.12 | 90.55±0.39 | 92.32±0.49 | 93.57±0.06 |
| FSTS | **87.02±0.25** | **89.86±0.19** | 91.11±0.38 | **92.83±0.03** | 94.09±0.20 |

For the KITTI Road experiments (Tables 1 and 2), the best results for each scenario are highlighted in bold. FSTS outperforms the competing methods, with performance gains being particularly pronounced in scenarios with smaller sample ratios.

For CIFAR-10 experiments, as shown in Table 3, the best result in each case is in bold. FSTS consistently outperforms most of baseline methods, especially in small ratio dataset settings. The

Table 2: Dice score on KITTI Road

| Method/Ratio | 10% | 20% | 30% | 40% | 50% |
|---|---|---|---|---|---|
| Random | 91.84±0.36 | 94.02±0.04 | 94.60±0.28 | 95.52±0.10 | **96.89±0.13** |
| Craig | 91.71±0.59 | 94.14±0.14 | **95.19±0.07** | 95.30±0.26 | 96.55±0.22 |
| Glisten | 91.63±0.54 | 93.08±0.31 | 94.44±0.25 | 95.22±0.11 | 96.02±0.53 |
| Graph Cut | 92.31±0.27 | 92.61±0.35 | 94.69±0.04 | 95.24±0.31 | 96.01±0.21 |
| Cal | 90.70±0.36 | 93.56±0.11 | 94.79±0.17 | 95.89±0.07 | 96.79±0.06 |
| Forgetting | 91.82±0.31 | 93.97±0.09 | 94.60±0.21 | 95.70±0.26 | 96.35±0.07 |
| FSTS | **92.51±0.20** | **94.23±0.13** | 94.93±0.23 | **95.94±0.07** | 96.67±0.11 |

Table 3: Test accuracy on CIFAR - 10

| Method/Ratio | 10% | 20% | 30% | 40% | 50% |
|---|---|---|---|---|---|
| Random | 85.17±0.34 | 87.83±0.23 | 89.47±0.42 | 90.41±0.15 | **91.33±0.15** |
| Craig | 83.00±0.81 | 86.72±0.72 | 88.41±0.42 | 89.41±0.34 | 90.39±0.37 |
| Glisten | 82.05±0.68 | 85.16±0.57 | 87.73±0.28 | 89.60±0.32 | 90.23±0.16 |
| Graph Cut | 83.20±0.51 | 85.96±0.24 | 87.42±0.14 | 88.85±0.33 | 90.02±0.34 |
| Cal | 82.77±0.40 | 84.30±0.43 | 86.06±0.15 | 87.66±0.30 | 88.37±0.34 |
| Forgetting | 83.85±0.41 | 87.24±0.50 | 88.48±0.34 | 89.59±0.31 | 90.83±0.03 |
| FSTS | **85.65±0.53** | **88.30±0.24** | **89.58±0.39** | **90.59±0.18** | 91.20±0.24 |

analysis of the validation loss curves in Fig.2(a)–(e) shows that the model trained on the selected samples in FSTS converges faster.

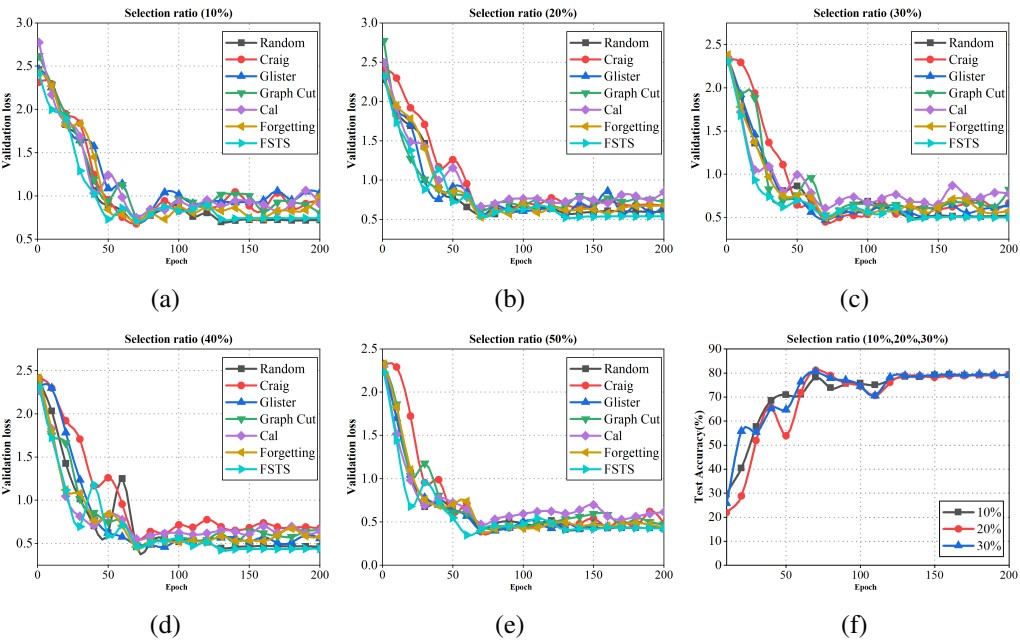

Figure 2: Validation loss curves of VGG-16 trained on different proportions of the dataset. (a) 10% of the dataset. (b) 20% of the dataset. (c) 30% of the dataset. (d) 40% of the dataset. (e) 50% of the dataset. (f) Test accuracy in the presence of label noise.

## 4.3 ROBUSTNESS EVALUATION

The KITTI road dataset not only includes roads, but also vehicles, buildings, trees, and obstructions that are closer to the real world. We use this dataset as the data foundation. To further evaluate the robustness of the proposed algorithm, we introduced label noise and image corruption into the CIFAR-10 dataset.

- Label Noise: We apply symmetric label noise (Lei et al., 2023) to 25% of the samples in each class, randomly flipping labels to simulate mislabeling. Then, core datasets of 10%, 20%, and 30% are selected using different methods and combined with the source data to train VGG-16.

- Image Corruption: We apply five types of image noise to 25% of samples per class: Gaussian noise, random occlusion, spots, salt and pepper noise, and shadow and light distortions. Fig. 3 in Appendix A.3 shows examples of the corrupted images. Again, core datasets of 10%, 20%, and 30% are selected and evaluated.

In the presence of label noise (see Table 4), the Graph Cut algorithm performs best due to its graph-based strategy for selecting representative central samples, followed closely by FSTS, which demonstrates strong robustness. In contrast, FSTS performs best under image corruption (see Table 5), particularly in small-data scenarios, confirming its reliability and adaptability in noisy conditions.

Table 4: Test accuracy on CIFAR-10 with label noise

| Method/Ratio | 10% | 20% | 30% |
| --- | --- | --- | --- |
| Random | 80.86±0.69 | 83.50±0.58 | 84.88±0.27 |
| Craig | 79.04±0.56 | 80.95±0.62 | 81.45±0.69 |
| Glister | 78.24±0.61 | 79.83±0.50 | 80.47±0.48 |
| Graph Cut | **82.59±0.75** | **85.44±0.16** | **85.36±0.48** |
| Cal | 76.28±1.31 | 80.93±0.93 | 83.91±0.29 |
| Forgetting | 80.90±0.55 | 83.46±0.31 | 84.77±0.31 |
| FSTS | 81.03±0.47 | 83.51±0.43 | 84.79±0.37 |

Table 5: Test accuracy on CIFAR-10 with image noise

| Method/Ratio | 10% | 20% | 30% |
| --- | --- | --- | --- |
| Random | 84.49±0.54 | 87.22±0.23 | 88.70±0.06 |
| Craig | 82.90±1.10 | 85.65±0.84 | 87.63±0.39 |
| Glister | 81.36±0.49 | 84.64±1.05 | 86.76±0.50 |
| Graph Cut | 83.08±0.38 | 85.87±0.48 | 86.87±0.41 |
| Cal | 82.46±0.75 | 84.42±1.01 | 86.49±0.32 |
| Forgetting | 83.49±1.11 | 85.82±0.52 | 87.97±0.63 |
| FSTS | **85.01±0.17** | **87.38±0.21** | **88.77±0.19** |

Furthermore, during label noise training, we observe that test accuracy peaks between epochs 60–70 and then declines and stabilizes for most of the methods, as illustrated in Fig. 2(f). This behavior is related to the evolution of the decision boundary (Lei et al., 2023). While noisy samples initially exert a strong influence on decision boundary formation, continued training enables the model to diminish their impact, leading to enhance generalization and overall robustness.

Table 6: Comparison of computational complexity and speed for selection methods

| Method | Core Steps | Time Complexity | Speed Rating |
| --- | --- | --- | --- |
| Random | Random index selection | $O(N)$ | Fastest |
| Cal | Feature extraction + nearest neighbor search | $O(N \cdot d + N \cdot k)$ | Fast |
| FSTS | Feature extraction + cosine distance sorting | $O(N \cdot T_{feat} + N \log N)$ | Medium |
| Craig | Gradient computation + greedy selection | $O(N \cdot d \cdot L + M \cdot N \cdot d)$ | Slow |
| Forgetting | Full training + forgetting counts | $O(E \cdot N \cdot T_{model})$ | Slow |
| Graph Cut | Similarity matrix construction + minimum cut | $O(N^2 \cdot d + N^{2.5})$ | Very Slow |
| Glister | Bi-level optimization + implicit differentiation | $O(K \cdot (M \cdot L + N) \cdot d)$ | Slowest |

Notions: $N$: Total number of samples in the dataset; $M$: Number of coreset samples; $d$: Feature dimension; $T_{feat}$: Time cost of feature extraction for one sample; $L$: Number of model parameters; $E$: Training epochs in the Forgetting algorithm; $T_{model}$: Time cost of forward/backward passes for one sample; $k$: Nearest neighbors in CAL($K \ll N$); $K$: Outer iterations in GLISTER;

## 4.4 TIME COMPLEXITY ANALYSIS

From the experimental results, it can be seen that the FSTS method balances computational efficiency and model performance. The experiments of CIFAR-10 classification and KITTI road segmentation show that its performance is superior to other selection methods.

## 5 CONCLUSION

In this paper, we proposes a data selection method based on FSTS for selecting high-quality data in deep learning image model training. This method enables users to select highly relevant core subsets from the target dataset, making more efficient use of the data and thus training deep neural networks more effectively, reducing storage, transmission, and transaction costs of the data exchange platform. Comparative experiments show that our method has performance and computational complexity advantages in building high-quality core datasets. The experiment also demonstrated the robustness of this method under conditions of image loss and label noise. In future work, we will demonstrate the effectiveness of this method in handling multimodal data.

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

# A APPENDIX

## A.1 CLOSEST, FARTHEST AND TWO-END METHODS

Here we introduce three different centroid similarity-based data selection methods— Closest, Farthest and Two-ends. For all methods, the value of $k$ denotes the number of samples to be selected per class from the target dataset, i.e., $k = \|z_j\| \times R$.

- Closest: Selects the samples closest to the class center in the target dataset, similar to the Herding method (Welling, 2009). The core dataset for each class is defined as:

$$\hat{\mathbb{D}} = \{D_{low}, \cdots, D_{upper}\}, \quad \text{where } low = 1, \ upper = k \quad (6)$$

- Farthest: Selects the samples farthest from the class center. The core dataset is

$$\hat{\mathbb{D}} = \{D_{low}, \cdots, D_{upper}\}, \quad \text{where } low = \|z_j\| - k, \ upper = \|z_j\| \quad (7)$$

- Two-ends: Combines the closest and farthest samples. The closest portion has $low_c = 1$, $upper_c = round(k/2)$, while the farthest portion has $low_f = \|z_j\| - trunc(k/2) + 1$, $upper_f = \|z_j\|$. The final core set is

$$\hat{\mathbb{D}} = \{D_{low_c}, \cdots, D_{upper_c}\} + \{D_{low_f}, \cdots, D_{upper_f}\} \quad (8)$$

## A.2 EXPERIMENTS WITH CLOSEST, FARTHEST AND TWO-ENDS

We conducted experiments using the Closest, Farthest, and Two-Ends methods on the KITTI Road dataset, the CIFAR-10 dataset, and robustness datasets with label noise and image corruption, and compared the results with the FSTS method. The experimental results show that, compared with the other three class-centroid-based selection methods, FSTS achieves better performance.

Table 7: IoU on KITTI Road

| Method/Ratio | 10% | 20% | 30% | 40% | 50% |
|---|---|---|---|---|---|
| Closest | 85.51±0.42 | 87.83±0.46 | 90.82±0.09 | 92.00±0.17 | 93.64±0.24 |
| Farthest | 85.87±0.53 | 88.99±0.46 | 90.87±0.09 | 92.52±0.10 | 93.47±0.79 |
| Two-ends | 86.35±0.24 | 87.21±0.73 | 89.46±0.48 | 92.17±0.11 | 93.74±0.12 |
| FSTS | **87.02±0.25** | **89.86±0.19** | **91.11±0.38** | **92.83±0.03** | **94.09±0.20** |

Table 8: Dice score on KITTI Road

| Method/Ratio | 10% | 20% | 30% | 40% | 50% |
|---|---|---|---|---|---|
| Closest | 91.63±0.26 | 93.05±0.30 | 94.84±0.05 | 95.49±0.12 | 96.45±0.12 |
| Farthest | 91.81±0.35 | 93.61±0.40 | 94.79±0.15 | 95.78±0.06 | 96.32±0.35 |
| Two-ends | 92.11±0.14 | 92.52±0.50 | 93.96±0.22 | 95.59±0.08 | 96.48±0.08 |
| FSTS | **92.51±0.20** | **94.23±0.13** | **94.93±0.23** | **95.94±0.07** | **96.67±0.11** |

Table 9: Test accuracy on CIFAR-10

| Method/Ratio | 10% | 20% | 30% | 40% | 50% |
|---|---|---|---|---|---|
| Closest | 84.89±0.41 | 87.18±0.19 | 88.59±0.40 | 89.64±0.13 | 90.63±0.25 |
| Farthest | 83.87±0.13 | 86.52±0.26 | 88.44±0.36 | 90.05±0.16 | 90.71±0.30 |
| Two-ends | 85.14±0.90 | 87.63±0.15 | 89.44±0.20 | 90.39±0.24 | 91.16±0.18 |
| FSTS | **85.65±0.53** | **88.30±0.24** | **89.58±0.39** | **90.59±0.18** | **91.20±0.24** |

Table 10: Test accuracy on CIFAR-10 with label noise

| Method/Ratio | 10% | 20% | 30% |
|---|---|---|---|
| Closest | 71.43±1.17 | 74.47±1.62 | 75.37±0.86 |
| Farthest | 81.02±0.30 | 83.16±0.26 | 84.26±0.22 |
| Two-ends | 76.62±2.25 | 76.77±0.96 | 80.03±0.72 |
| FSTS | **81.03±0.47** | **83.51±0.43** | **84.79±0.37** |

Table 11: Test accuracy on CIFAR-10 with image noise

| Method/Ratio | 10% | 20% | 30% |
|---|---|---|---|
| Closest | 84.20±0.40 | 86.54±0.28 | 88.14±0.24 |
| Farthest | 84.10±0.33 | 86.55±0.40 | 88.34±0.16 |
| Two-ends | 84.28±0.63 | 86.99±0.65 | 88.67±0.16 |
| FSTS | **85.01±0.17** | **87.38±0.21** | **88.77±0.19** |

## A.3 IMAGE CORRUPTION

In the paper, we employ five type of noises for the image corruption, i.e. gaussian noise, random occlusion, spots, salt and pepper noise, and shadow and light distortions. The noise is shown in Fig.3.

| Gaussian | Random Occlusion | Spots | Salt and Pepper | Shadow and Light |
|---|---|---|---|---|

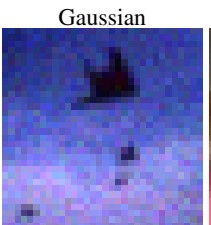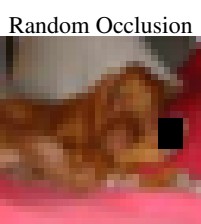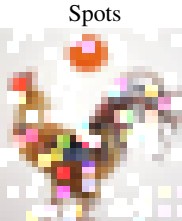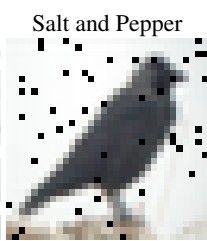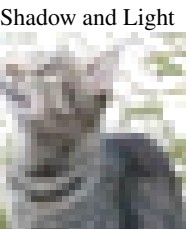

Figure 3: Noise processing on the target dataset.

