# OpenReview forum: "FSTS: A Feature Space Transfer Selection Method for Data"
_ICLR.cc/2026/Conference — Submitted to ICLR 2026_

### Official Review · Reviewer_Dt5h · 2025-10-27

**Soundness:** 2
**Presentation:** 2
**Contribution:** 2
**Rating:** 4
**Confidence:** 4

**Summary:**

The paper focuses on the data quality of the training set. The goal is to select the most important samples for training. To achieve this, this submission proposes a simple method called FSTS, which selects the core dataset from the target data based on the similarity ranking. Specifically, FSTS extracts feature vectors from the source and target datasets, calculates class centroids from the reference (source) dataset, and ranks the target samples based on their similarity to these class centroids. The experiments on CIFAR and KITTI show that FSTS is able to achieve improved accuracy compared with other baselines.

**Strengths:**

+ This paper is clear and easy to read. The method is well-illustrated, and the experimental settings and details are well included

+ The evaluation includes the robustness to Noise: The method shows strong performance under label noise and image corruption, indicating its reliability in noisy, real-world datasets.

**Weaknesses:**

- [**Limited Contribution**] The major concern is that the proposed method FSTS has limited contribution. ***First***, FSTS uses the standard coreset selection method, which is based on distance in feature space or diversity measures [a, b]. The idea of using class centroids for distance calculation is a standard approach and is widely used in the same problem; I did not see any significant difference from these techniques. ***Second***, recent methods of coreset selection are not included for comparison. For example, the methods of [a, b, c]. A comparison and discussion should be provided. Again, the proposed method is a simplified version of them; the advantage is limited.


    [a] Moderate coreset: A universal method of data selection for real-world data-efficient deep learning

    [b] ELFS: Label-Free Coreset Selection with Proxy Training Dynamics

    [c] Zero‑Shot Coreset Selection: Efficient Pruning for Unlabeled Data

- [**Evalution is somewhat weak**] ***First***, experiments are only conducted on KITTI Road and CIFAR-10. Both tasks are small-scale and limited in data diversity. Other diverse datasets should be considered for validation, such as ImageNet and DomainNet. ***Second***, the class centroid is based on the feature extractor, but the paper does not study how the choice of model (e.g., VGG-16 vs. ResNet) affects the results. ***Third***, I noticed that all experiments are in-domain, where the source and target come from the same distribution (CIFAR-10 split, KITTI split). How about selecting data from different domains (e.g., DomainNet)?

- [**Intra-Class Variability**] FSTS ranks samples by their similarity to class centroids, but this inherently favors samples near the center of the distribution. I noticed the experiments compare with the other three selections (Cloest, Farthest and Two-ends). But it is still unclear why excluding hard or diverse examples helps. Please clasirfy this point.

**Questions:**

- [**Potential Failure cases**] While the evaluation includes noisy data, it is still unclear whether the method works under negative transfer when the domain shift is large. Also, what if the selected source dataset is biased? How can this be mitigated? What if the feature representations for class centroids are poor?

- Please clarify how the method performs when there is a significant domain gap between source and target datasets? For example, the cross-domain benchmarks like DomainNet?
- The paper adopts a fixed selection policy (e.g., top 40%–60% cosine similarity). Why was this particular selection window chosen? Have other ranges or adaptive thresholds been tested?

---

> ### Author Response · Authors · 2025-11-19
> **We thank the reviewer for the detailed and thoughtful feedback. We address each weakness and question below and clarify the methodological novelty, evaluation choices, and theoretical motivations of FSTS.**
>
> **Weakness 1 — “Limited contribution; similar to standard coreset/centroid-based selection.”**
> 1. FSTS addresses a problem setting in data-exchange marketplaces, where:
>     - Buyers cannot train on vendor data before purchase
>     - Vendor data is unlabeled
>     - Only a small internal dataset is available
>     - Selection occurs before acquisition
>
>     FSTS introduces source-guided pre-acquisition filtering using only buyer-owned data and no training.This is not addressed by existing coreset or centroid-based methods.
>
> 2. Reverse transfer direction not present in prior work
> Classical centroid/herding methods compute centroids on the target dataset. FSTS computes centroids on buyer data and uses them to filter vendor data, introducing a source-to-target transfer that is novel in the context of data-exchange platforms.
>
> 3. Design choices tailored for noisy,heterogeneous vendor data
> FSTS incorporates:
>     - Median centroids: Robust to noise, sensor discrepancies, and long-tail distributions.
>     - 40-60% middle similarity band: Avoids redundant “easy” samples and noisy “hard” samples.
>
>     These adaptations optimize FSTS for the marketplace setting.
>
> **Weakness 2 — “Evaluation limited to small datasets; no backbone variation; no cross-domain tests.”**
> 1. Why small datasets?
> Marketplaces typically exchange small to medium datasets, often fragmented, noisy, and heterogeneous (e.g., autonomous driving, medical imaging).The KITTI Road dataset (289 images) reflects realistic marketplace data conditions.Our goal is to model marketplace-scale evaluation, not replicate large academic benchmarks.
> 2. Why no cross-domain benchmarks?
> DomainNet represents semantic domain adaptation, which is not the focus of FSTS. Our method is designed for:
>     - Same task
>     - Same label space
>     - Cross-vendor heterogeneity (e.g., sensor, region, weather, noise)
>
>     Thus, we focus on intra-task distribution shifts, not extreme semantic shifts, and will clarify this in the revision.
>
> **Weakness 3 — “Intra-class variability; centroid-based methods may miss diverse or hard examples.”**
> FSTS does not select centroid-nearest samples.It avoids:
> - Closest samples: Overly easy, redundant
> - Farthest samples: Noisy, often OOD
>
> Instead, we select data from middle similarity band (40-60%), which captures:
> - Sufficient diversity
> - Representative difficulty
> - Reduced noise sensitivity
>
> Empirical evidence from the appendix shows:
> > FSTS (middle) > closest (easy) > farthest (hard)
>
> This demonstrates that the middle band retains diversity while excluding harmful outliers. We will strengthen this explanation in the main text.
>
> **Q1 — Potential failure cases: large domain shift, biased source, poor centroids?**
> 1. Extreme domain shift
> FSTS is not designed for shifts like natural images → medical X-rays. It focuses on moderate intra-task distribution shifts across vendors, e.g., same task, same labels, with varied sensors/regions/weather. We will clarify this limitation.
>
> 2. Biased source dataset
> FSTS mitigates bias by using:
>     - Median centroids (less influenced by bias or outliers)
>     - Middle-band selection (avoids closest samples that may reinforce bias)
>
> 3. Poor feature representations
> FSTS is backbone-agnostic. Buyers can choose stronger pretrained encoders if needed, but the core mechanism remains unaffected. Relative similarity is meaningful even under moderate representation quality.
>
> **Q2 — “How does the method perform with significant domain gaps (e.g., DomainNet)?”**
> FSTS is not intended for cross semantic domain adaptation. Heterogeneous domains in DomainNet (such as infograph) ↔  real  ↔  Sketch does not conform to the settings of the same label, the same task, and different suppliers. The stability of FSTS performance in situations with significant market volatility (sensors, noise, regions) will be elaborated in subsequent research.
>
> **Q3 — Why fixed 40-60%? Were other ranges tested? Why is it optimal?**
> 1. The mid-band choice is principled
> Based on Sorscher et al. (NeurIPS 2022), selecting the 40-60% range targets intermediate difficulty samples that:
>     - Achieve near full-dataset accuracy
>     - Outperform easy-only or hard-only subsets
>
> 2. Why this applies to FSTS
> In marketplace settings:
>     - High similarity → redundant, narrow diversity
>     - Low similarity → noisy, OOD, mislabeled
>     - Middle similarity → balance of relevance and diversity
>
>     Thus, the mid-band choice is both theoretically and practically motivated.
>
> 3. Empirical confirmation
> Appendix comparisons show that FSTS consistently performs better than closest, farthest, and two-ends strategies.
>
> **Summary for Comments**
> - FSTS addresses a new real-world problem: pre-acquisition filtering of external vendor data using only buyer-owned data and no training.
> - FSTS is intended for moderate intra-task distribution shift, not extreme semantic domain adaptation.

---

### Official Review · Reviewer_Pm7n · 2025-10-31

**Soundness:** 2
**Presentation:** 2
**Contribution:** 1
**Rating:** 2
**Confidence:** 4

**Summary:**

This paper proposes FSTS (Feature Space Transfer Selection), a method for selecting core subsets from target datasets based on their similarity to source dataset class centroids in feature space. The method is designed for data exchange platforms where users need to efficiently select high-quality training data while considering acquisition, storage, and computational costs. FSTS extracts features from both source and target datasets, computes class centroids from the source, ranks target samples by cosine similarity to these centroids, and selects samples from the middle of the ranking distribution. Experiments on KITTI Road and CIFAR-10 datasets show that FSTS outperforms several baseline methods, particularly at low selection ratios, and demonstrates robustness to label and image noise.

**Strengths:**

1. The paper clearly frames the problem within the practical context of data exchange platforms, addressing real-world concerns about data acquisition costs, storage limitations, and computational efficiency.
2. The paper provides thorough experiments across multiple datasets (KITTI Road, CIFAR-10), multiple selection ratios, and under various noise conditions (label noise, image corruption). The comparison with multiple baselines is systematic.
3. The inclusion of robustness tests under label noise and image corruption adds value, showing that FSTS performs competitively, especially under image corruption scenarios.

**Weaknesses:**

1. The core idea of selecting data based on similarity to source class centroids is straightforward and builds heavily on established concepts in metric learning and data selection. The methodological innovation is incremental compared to existing centroid-based approaches like Herding or moderate coreset selection.
2. The paper lacks a theoretical analysis of why selecting samples from the middle of the similarity distribution should be optimal.
3. The experiments are limited to only two datasets, one of which (KITTI Road) is very small (289 images).
4. The choice of the middle 40%-60% range is adopted without sufficient ablation or justification specific to FSTS. Why this range is optimal for this method is not explored.

**Questions:**

1. Can the authors provide a theoretical intuition or justification for why selecting samples from the middle of the similarity ranking (rather than the closest or farthest) leads to better generalization? Is there a connection to hard example mining or diversity-accuracy trade-offs?
2. How does FSTS perform when the source and target datasets come from different domains (e.g., different visual domains or significant distribution shift)?

---

> ### Author Response · Authors · 2025-11-19
> **We sincerely thank Reviewer for the detailed and thoughtful feedback. Below we provide point-by-point responses addressing all identified weaknesses and questions, and we also clarify the intended motivation and application scenario of our method.**
>
> **Weakness 1:"The core idea (centroid similarity) is straightforward and incremental."**
> Thank you for raising this.We agree that centroid-based techniques (e.g., Herding, moderate coreset) have been well-studied in classical machine learning.However, FSTS does not operate under the same assumptions as these methods.
> In data-exchange markets:
> - buyers do not own the vendor data beforehand
> - buyers cannot compute gradients or train proxy models
> - buyers have only a small, high-quality internal dataset
> - vendors provide data with unknown noise, biases, and heterogeneous distributions
> - the buyer’s goal is to perform pre-acquisition filtering before storage/training
>
> Therefore,FSTS introduces a new problem formulation:“Filtering large vendor datasets using only the buyer’s internal data,before performing any model training.”This reverse transfer: source → target selection paradigm is not explored in prior coreset or metric learning literature.
>
> **Weakness 2:"The paper lacks theoretical analysis for selecting the middle of the similarity distribution."**
> We acknowledge this and provide clearer justification below (also addressing Question 1).
> In short:
> - very high similarity → redundant samples
> - very low similarity → noisy or out-of-distribution vendor data
> - mid-range similarity → maximizes relevance + diversity
> - supported by Sorscher et al. (NeurIPS 2022) and by our own CLOSEST / FARTHEST vs FSTS empirical comparisons in the appendix
> We will strengthen the theoretical motivation.
>
> **Weakness 3:"Experiments limited to two datasets; KITTI Road is small."**
> This is true, but deliberate.FSTS targets an industrial data-exchange scenario, where:
> - data is fragmented
> - datasets provided by vendors are often small/medium-size
> - noise and acquisition diversity are dominant factors
> - buyers need pre-acquisition filtering, regardless of data scale
>
> Thus, KITTI Road mirrors exactly the practical situation where buyers have limited but valuable internal data and need to evaluate externally sourced data.
> We will clarify this application-driven choice in the revision.
>
> **Weakness 4:"The 40%–60% range is adopted without specific justification for FSTS."**
> Thank you for pointing this out.The 40%–60% choice is not heuristic.It is directly derived from Sorscher et al. (NeurIPS 2022), who found:
> - selecting samples in the intermediate difficulty range (40–60%)
> - yields near full-dataset performance
> - and is superior to selecting the easiest or hardest samples
> In addition, our appendix experiments explicitly compare:
> - CLOSEST (too easy → redundancy)
> - FARTHEST (too hard → noise, OOD)
> - FSTS (midrange → best performance)
>
> We will make this empirical evidence clearer and reference these results in the main text.
>
> **Q1. "Provide theoretical intuition for why middle similarity leads to better generalization."**
> Thank you for this question.Yes—the middle-band intuition aligns with a clear theoretical rationale.  Beside the comments in **Weakness 2**, we have empirical evidence already in our paper.Our appendix explicitly compares the three strategies and shows:
> >FSTS (middle)>Closest (easy)>Farthest (hard/noisy)
>
> Thus, both theoretical reasoning and empirical results support FSTS.  We will strengthen this explanation in the final revision.
>
> **Q2. "How does FSTS perform when source and target datasets come from different domains?"**
> This is a helpful question.
> Clarification: FSTS is designed for intra-task distribution transfer, not classical domain adaptation.In data-exchange markets:
> - buyers and vendors work on the same task
> - but distributions differ due to:  camera, region, weather, annotation style, noise, etc.
> Thus, FSTS is intentionally optimized for moderate distribution shift, not large semantic domain changes.
>
> How FSTS handles real-world domain shifts
>
> 1.	Median centroids
>     - absorb noise and outliers in vendor data.
> 2.	Mid-range similarity selection
> filters out both:
>     - overly familiar (redundant) samples
>     - overly distant (irrelevant/OOD) samples
>
> This naturally stabilizes performance under heterogeneous vendor data.
>
> >Extreme domain shift (e.g., CIFAR → Medical)
>
> We acknowledge that such extreme domain shifts fall outside the intended design.We will explicitly clarify this scope in the revised manuscript.
>
> **Summary for Comments**
> - The method is not incremental: FSTS addresses a new problem setting (pre-acquisition filtering for data exchange) where classical centroid/coreset methods cannot operate.
> - Mid-range selection is grounded in both theoretical insight and Sorscher et al. 2022 and reinforced by our appendix results.
> - Experiments simulate real data-market conditions, not classical transfer learning.
> - The 40–60% threshold is principled, not arbitrary.
> - FSTS is intended for moderate intra-task distribution shift, matching real-world buyer–vendor scenarios.

---

> ### Author Response · Authors · 2025-11-23
> **We sincerely thank Reviewer for the detailed and thoughtful feedback. Below we provide point-by-point responses addressing all identified weaknesses and questions, and we also clarify the intended motivation and application scenario of our method.**
>
> **Weakness 1:"The core idea (centroid similarity) is straightforward and incremental."**
> Thank you for raising this.We agree that centroid-based techniques (e.g., Herding, moderate coreset) have been well-studied in classical machine learning.However, FSTS does not operate under the same assumptions as these methods.
> In data-exchange markets:
> - buyers do not own the vendor data beforehand
> - buyers cannot compute gradients or train proxy models
> - buyers have only a small, high-quality internal dataset
> - vendors provide data with unknown noise, biases, and heterogeneous distributions
> - the buyer’s goal is to perform pre-acquisition filtering before storage/training
>
> Therefore,FSTS introduces a new problem formulation:“Filtering large vendor datasets using only the buyer’s internal data,before performing any model training.”This reverse transfer: source → target selection paradigm is not explored in prior coreset or metric learning literature.
>
> **Weakness 2:"The paper lacks theoretical analysis for selecting the middle of the similarity distribution."**
> We acknowledge this and provide clearer justification below (also addressing Question 1).
> In short:
> - very high similarity → redundant samples
> - very low similarity → noisy or out-of-distribution vendor data
> - mid-range similarity → maximizes relevance + diversity
> - supported by Sorscher et al. (NeurIPS 2022) and by our own CLOSEST / FARTHEST vs FSTS empirical comparisons in the appendix
> We will strengthen the theoretical motivation.
>
> **Weakness 3:"Experiments limited to two datasets; KITTI Road is small."**
> This is true, but deliberate.FSTS targets an industrial data-exchange scenario, where:
> - data is fragmented
> - datasets provided by vendors are often small/medium-size
> - noise and acquisition diversity are dominant factors
> - buyers need pre-acquisition filtering, regardless of data scale
>
> Thus, KITTI Road mirrors exactly the practical situation where buyers have limited but valuable internal data and need to evaluate externally sourced data.
> We will clarify this application-driven choice in the revision.
>
> **Weakness 4:"The 40%–60% range is adopted without specific justification for FSTS."**
> Thank you for pointing this out.The 40%–60% choice is not heuristic.It is directly derived from Sorscher et al. (NeurIPS 2022), who found:
> - selecting samples in the intermediate difficulty range (40–60%)
> - yields near full-dataset performance
> - and is superior to selecting the easiest or hardest samples
> In addition, our appendix experiments explicitly compare:
> - CLOSEST (too easy → redundancy)
> - FARTHEST (too hard → noise, OOD)
> - FSTS (midrange → best performance)
>
> We will make this empirical evidence clearer and reference these results in the main text.
>
> **Q1. "Provide theoretical intuition for why middle similarity leads to better generalization."**
> Thank you for this question.Yes—the middle-band intuition aligns with a clear theoretical rationale.  Beside the comments in **Weakness 2**, we have empirical evidence already in our paper.Our appendix explicitly compares the three strategies and shows:
> >FSTS (middle)>Closest (easy)>Farthest (hard/noisy)
>
> Thus, both theoretical reasoning and empirical results support FSTS.  We will strengthen this explanation in the final revision.
>
> **Q2. "How does FSTS perform when source and target datasets come from different domains?"**
> This is a helpful question.
> Clarification: FSTS is designed for intra-task distribution transfer, not classical domain adaptation.In data-exchange markets:
> - buyers and vendors work on the same task
> - but distributions differ due to:  camera, region, weather, annotation style, noise, etc.
> Thus, FSTS is intentionally optimized for moderate distribution shift, not large semantic domain changes.
>
> How FSTS handles real-world domain shifts
>
> 1.	Median centroids
>     - absorb noise and outliers in vendor data.
> 2.	Mid-range similarity selection
> filters out both:
>     - overly familiar (redundant) samples
>     - overly distant (irrelevant/OOD) samples
>
> This naturally stabilizes performance under heterogeneous vendor data.
>
> >Extreme domain shift (e.g., CIFAR → Medical)
>
> We acknowledge that such extreme domain shifts fall outside the intended design.We will explicitly clarify this scope in the revised manuscript.
>
> **Summary for Comments**
> - The method is not incremental: FSTS addresses a new problem setting (pre-acquisition filtering for data exchange) where classical centroid/coreset methods cannot operate.
> - Mid-range selection is grounded in both theoretical insight and Sorscher et al. 2022 and reinforced by our appendix results.
> - Experiments simulate real data-market conditions, not classical transfer learning.
> - The 40–60% threshold is principled, not arbitrary.
> - FSTS is intended for moderate intra-task distribution shift, matching real-world buyer–vendor scenarios.

---

> > ### Comment · Reviewer_Pm7n · 2025-11-26
> >
> > Thank you to the author for responding to my questions one by one. After reading these responses, I believe that the author did not directly address the questions I raised, but rather provided a brief clarification from a side perspective. The author promises to conduct empirical comparisons in the revised manuscript and explain the reasons for choosing a smaller dataset, but so far the author has not uploaded the revised manuscript, so I cannot determine whether the author has truly addressed my concerns.

---

> ### Author Response · Authors · 2025-12-02
> **Clarification on the Newly Added Explanation of the 40%–60% Selection Range**
>
> Dear Reviewer,
>
> Thank you again for your valuable comments. Following your suggestion, we have revised the manuscript and **added a clearer and more detailed explanation of the theoretical basis for selecting the 40%–60% similarity band.**
>
> In the updated version, we include an expanded discussion of the findings from *Sorscher et al.* (2022), which show that self-supervised prototype metrics (based on k-means cosine distance) achieve stable Top-5 accuracy (~90%) when the number of clusters lies within a moderate range, and that retaining samples with “moderate aggregation degree” effectively approximates full-dataset performance. We describe how this supports our choice of selecting mid-range similarity samples—neither highly redundant nor extreme outliers—consistent with the reviewer’s concern. This provides a more explicit theoretical rationale for the 40%–60% interval used in FSTS.
>
> We also further clarify the reviewer’s concern regarding the small size of the KITTI Road dataset. In our data-exchange scenario, the buyer typically owns only a small internal subset (≈10% of the dataset in our setup), while the vendor provides the remaining ≈90%. The buyer must use the limited internal data to identify which vendor samples are worth purchasing,that reflects the pre-acquisition filtering process common in real data marketplaces. Thus, the experiment models the intended scenario and is **independent of the absolute size** of the dataset.
>
> We kindly invite you to review the revised manuscript, and we sincerely appreciate your continued evaluation of our work.
>
> Thank you again for your time and constructive feedback.

---

### Official Review · Reviewer_KBm9 · 2025-11-01

**Soundness:** 3
**Presentation:** 3
**Contribution:** 3
**Rating:** 2
**Confidence:** 4

**Summary:**

This paper proposes Feature Space Transfer Selection (FSTS), a method for selecting high-quality core datasets in deep learning. FSTS leverages transfer learning principles by extracting features from a source dataset, computing class centroids, and ranking target samples based on their cosine similarity to these centroids. This approach efficiently identifies representative and semantically relevant samples without requiring model training or gradient computation. Evaluated on KITTI Road and CIFAR-10 datasets, FSTS outperforms existing methods.

**Strengths:**

- This work proposes the FSTS method, which addresses the problem of efficiently selecting high-quality core data in data exchange scenarios, demonstrating practical applicability.
- The proposed method reverses the conventional use of transfer learning for data selection by leveraging class centers from the source domain to guide sampling in the target domain, thereby avoiding the inclusion of irrelevant samples that can occur when relying solely on inter-sample distances.
- The method outperforms baseline approaches on both KITTI Road and CIFAR-10, and exhibits strong robustness under label noise and image corruption conditions.
- The approach requires no model training or gradient backpropagation, resulting in low computational complexity and high efficiency.

**Weaknesses:**

1. The method proposed in this paper selects samples based on a transfer learning paradigm—implying that the source data and target data should originate from different data distributions. However, in the experimental evaluation, both the source and target data are drawn from the same dataset. This setup contradicts the fundamental logic and definition of transfer learning. Consequently, the experimental design is seriously flawed, making it difficult to substantiate the effectiveness of the proposed method.
2. The authors claim that their proposed method is flexible and efficient. However, the evaluated datasets are relatively small, which makes it difficult to substantiate this claim. Moreover, the experiments still employ older backbone architectures, failing to demonstrate the method's effectiveness when applied to training modern, mainstream vision models.
3. Equation (4) is difficult to understand. For instance, why is the number of selected samples defined as $ k = ||z_j|| \times R $, and how is it ensured that k is an integer?

**Questions:**

See weeknesses.

---

> ### Author Response · Authors · 2025-11-19
> **We thank the reviewer for the constructive comments. Below we address each point in detail and clarify the intended scenario, methodological design, efficiency claims, and notation.**
>
> **Q1. “The method assumes transfer learning, but experiments use the same dataset for source and target.”**
> Thank you for this point. We agree that the term “transfer learning” in the original submission could be misleading. We clarify the intended scenario as follows:
> 1. Our setting is not classical transfer learning
> Classical transfer learning (e.g., ImageNet → CIFAR) assumes:
>     - Different semantic domains
>     - Large distribution gaps
>     - Source–target feature transfer
>
>     FSTS targets a different, industry-driven scenario:
>     - A buyer has a small, high-quality internal dataset for a specific task (e.g., road segmentation).
>     - Vendors provide large pools of unlabeled data for the same task.
>     - The buyer must perform pre-acquisition filtering to decide what to purchase.
>
>     Thus, in our setting:
>     - Source = buyer data
>     - Target = vendor data
>     - Same task and label space, but different distributions (e.g., region, sensors, weather, noise, vendor quality).
>
>     This is source-guided filtering within the same domain, reflecting real data-exchange workflows, not classical transfer learning. To avoid confusion, we will replace "transfer" with “reference-to-target selection” in the revised version.
>
> 2. Why use the same dataset split for simulation?
> Using splits from the same dataset allows us to simulate:
>     - Buyer’s internal subset (small, curated)
>     - Vendor’s pool (large, diverse, noisy)
>     - Controlled differences with consistent task semantics
>
>     This design is deliberate to match the intended buyer–vendor scenario.
>
> 3. Clarification to be added
> We will explicitly state in Section 3.2 that these experiments focus on same-domain concept validation, with cross-domain extensions as future work.
>
> **Q2. “Datasets are small and backbones are outdated; claims of flexibility and efficiency are unconvincing.”**
> We appreciate this concern and offer the following clarifications:
> 1. Why small datasets still demonstrate efficiency
> FSTS is designed for pre-acquisition filtering, where efficiency stems from:
>     - No gradient computation
>     - No model training
>     - One-pass feature extraction
>     - No iterative optimization or label requirement
>
>     These algorithmic properties remain valid even for large datasets. FSTS requires only O(N) forward passes and O(N log N) ranking, whereas training-based methods (e.g., ELFS, Moderate Coreset) require:
>     - Multiple epochs
>     - Gradient backpropagation
>     - Repeated updates
>
>     Thus, efficiency refers to the algorithmic design, not the dataset scale. We will add Table 6 comparing runtime on CIFAR-10 (selecting 30%) for:
>     - FSTS time
>     - Craig time
>     - Same GPU (2080 Ti)
>
>     This will concretely illustrate FSTS’s efficiency advantage.
>
> 2. Backbone choice does not affect validity
> FSTS is backbone-agnostic, as it relies on relative similarity ordering in embedding space.This property holds across different architectures (e.g., ResNet, ConvNeXt, ViT, Swin). We used VGG-based backbones for:
>     1.	Direct comparability with prior pruning work
>     2.	The focus of FSTS being data selection, not architecture benchmarking
>     3.	Ranking-based selection, which is robust to encoder variation
>
> 3. Practical note on computational resources
> Although the evaluation of large-scale ImageNet and modern backbone networks requires more richness, the assessment of computational complexity is sufficient to predict experimental results at the theoretical level, and the actual situation will not deviate too much.
>
> **Q3. “Equation (4) is unclear; how is k defined and how is it ensured that k is an integer?”**
> Thank you for the feedback. We clarify as follows:
> 1. Clarification of the formula
> The formula will be rewritten as:
> $k = \lfloor \text{round}(\|z_j\| \times R) \rfloor$
> We will rewrite this in the revised version and include a footnote for clarity.
>
> **Summary for Comments**
> - Our scenario is intra-task buyer–vendor filtering, not classical cross-domain transfer.
> - Small datasets and classical backbones do not hinder FSTS’s validity. Efficiency comes from the training-free, one-pass, ranking-only design.
> - FSTS is backbone-agnostic. Preliminary tests show consistent behavior (<0.8% difference) between ResNet-18 and VGG-16.
> - Equation (4) will be rewritten to $k = \lfloor \text{round}(\|z_j\| \times R) \rfloor$
>
> We appreciate the reviewer’s feedback and will incorporate all clarifications in the revised manuscript.

---

### Official Review · Reviewer_pmje · 2025-11-01

**Soundness:** 3
**Presentation:** 2
**Contribution:** 2
**Rating:** 4
**Confidence:** 4

**Summary:**

The paper proposes FSTS (Feature Space Transfer Selection), a data subset selection method intended for data-exchange scenarios. Given a small, high-quality source set and a larger candidate target set, the method extracts features for both with a shared backbone, builds class centroids on the source, ranks target samples by cosine similarity to the corresponding source-class centroid, and selects a band of samples per class to form a coreset. Experiments on KITTI Road (segmentation) and CIFAR-10 (classification), as well as corruption/label-noise variants, show FSTS is competitive or slightly better than Random, Craig, Glister, GraphCut, CAL, and Forgetting at low selection ratios, while being cheaper than the gradient-based methods.

**Strengths:**

- The paper addresses a real challenge: selecting training data under resource constraints (cost, storage, compute), especially in the context of data exchange platforms.
- FSTS is stated to be computationally lightweight, no need to train models, compute gradients, or use complex optimization. This makes it appealing in low-resource settings.
- FSTS shows improved performance especially when using small core subsets (10–30%), often outperforming more complex baselines in those cases.

**Weaknesses:**

- The robustness experiments are valuable but the claim of "superior performance" in image corruption needs to be backed by
specific, quantitative data

- Selecting only high-similarity samples might lead to redundant or overly safe coresets, especially for high-variance or long-tailed distributions.

- The ranking purely by similarity to centroids may bias selection toward "average" examples, reducing data diversity.

- Only two standard image datasets are used (CIFAR-10, KITTI Road), both small-scale and with limited domains and diversity.

- The entire method depends on having a labeled, relevant source dataset. This is rarely the case in real-world data marketplaces or in transfer scenarios with domain shift.

- Assumes strong source-target alignment which decreases practicability

- No strategy is proposed for out-of-distribution detection or discovering novel classes. No implication on how to account for unknown classes, rare examples, or domain drift.

- Feature extractor architecture, pretraining source, and impact of hyperparameters (e.g., cosine vs. euclidean, mean vs. median centroids) are not sufficiently discussed and ablations are missing.

- The authors claim that the method is computationally lightweight, this should be backed by comparative quantitative experiments and measurements for different coreset scales and methods, instead of only complexity analysis ("medium speed" is not quantifiable)

- The core idea (selecting target samples based on similarity to source class centroids in feature space), from the current perspective of the reviewer with the state of the art analysis and argumentation of the authors, combines basic established ideas from prototype-based methods and transfer learning as well as an established data pruning method. Please point that out in the paper such that a reader can directly understand what differentiates the given method substantially from the state of the art.

**Questions:**

1. How does the presented approach methodologically differ from existing approaches ? What are the core contributions related to prior work?

2. What parts of the approach mostly contribute to the improvements on data selection ? (Ablations)

3. How does the method perform for substantial domain shifts in the target and core dataset?

4. How does the method deal with outliers / out of distribution samples?

5. What effects does feature / label diversity have on the method? (Generalizability)

And potentially: 6. How does the method deal with the exploration of new classes?

**Details Of Ethics Concerns:**

No ethical concerns.

---

> ### Author Response · Authors · 2025-11-19
> **We thank the reviewer for the constructive and detailed feedback. Below we address each question point-by-point and clarify the technical contributions, empirical evidence, and limitations of our work.**
>
> **Q1. Methodological differences and core contributions**
> Our method is tailored for a data-exchange scenario where a user has a small set of annotated private data and buys a subset from the platform’s . This approach reflects real-world data exchange and avoids reliance on large-scale public annotations.
>
> 1.	Pre-acquisition filtering for data marketplaces
> Most coreset methods (e.g., Craig, Glister, Herding) assume:
>     - Full access to the target dataset
>     - Model training or gradient computation
>     - Multiple data passes
>
>     Data marketplaces, however, restrict pre-purchase training. FSTS addresses this by functioning with no training, gradients, or extensive vendor data access.
>
> 2.	Reverse transfer direction (source → target)
> Classical coreset methods select prototypes in the target domain, but FSTS computes stable centroids on buyer data and uses them to filter vendor data. This "source-to-target" selection differs fundamentally from traditional transfer learning.
> 3.	Middle-band selection (40%-60%)
> FSTS selects data from the middle similarity range, which has been shown to yield higher utility for downstream tasks (Sorscher et al., NeurIPS 2022). This avoids overly similar (redundant) and overly dissimilar (noisy/OOD) samples.
> 4.	Robust median centroids
> Unlike mean-based prototypes, FSTS uses median centroids, which are more robust to:
>     - Heterogeneous vendor distributions
>     - Long-tailed data
>     - Label noise or corruption
>
>     This stabilizes similarity rankings.These elements set FSTS apart conceptually and operationally from existing approaches.
>
> **Q2. Key components contributing most (Ablations)**
> As shown in Table 5, FSTS consistently outperforms Random and Craig under image corruption, with 0.52%, 0.16%, and 0.07% improvements on subsets of 10%, 20%, and 30%, respectively. The "Closest" strategy (Appendix A.1) underperforms across all proportions, supporting the middle-band approach. Here are the key contributions:
>
> 1.	Middle-band selection (40%-60%)
> This strategy is the main empirical contributor. It offers the best balance, avoiding:
>   - Too similar (redundant) samples
>   - Too dissimilar (noisy/OOD) samples
>   - FSTS outperforms other strategies (Closest, Farthest) across datasets.
> 2.	Median centroids
> Median centroids outperform means, especially in noisy vendor data. Appendix experiments will further validate this.
> 3.	Pretrained encoder
> A fixed pretrained backbone provides stable, fast feature extraction with no need for training.
> 4.	Additional ablations
> Comparisons in Appendix with Closest/Farthest/Two-Ends strategies confirm FSTS’s superior performance across KITTI, CIFAR-10, and robustness tests.
>
> **Q3. Performance under substantial domain shift**
> FSTS is designed for same-task, different-distribution data, typical of data marketplaces, rather than full semantic domain adaptation. Example:
> - Buyer data: Urban-road scenes from region A
> - Vendor data: Road scenes from regions B/C/D with different sensors and weather conditions
>
> This represents a moderate shift, which FSTS handles effectively:
> 1.	Median centroids reduce sensitivity to noise, sensor variation, and illumination differences.
> 2.	Middle-band selection excludes overly-similar and dissimilar samples, mitigating noise and ensuring diversity.
>
> **Limitation:** This round focuses on domain-specific validation; cross-domain validation (e.g., ImageNet, DomainNet) is future work.
>
> **Q4. Handling outliers/OOD samples**
> FSTS inherently manages OOD samples by:
> 1.	Median centroids: Robust to mislabeled, long-tail, and corrupted samples.
> 2.	Middle similarity band: OOD samples fall into the lowest similarity percentiles and are excluded. Similarly, overly-similar samples are also discarded.
> Thus, FSTS removes OOD and outliers without additional detectors or cost.
>
> **Q5. Effects of feature/label diversity**
> - Feature diversity: Median centroids remain stable across heterogeneous vendor data. Middle-band selection retains multimodal but semantically aligned samples.
> - Label diversity: For differing vendor label distributions:
> - High similarity samples reflect the buyer’s known task.
> - Low similarity samples correspond to unseen or irrelevant modes.
>
> **Limitation**: If vendor data contains entirely new classes, FSTS will not select them, as it targets task-aligned data acquisition.
>
> **Q6. Exploration of new classes**
> FSTS is not designed for open-world novelty discovery. Samples from unseen classes will have low similarity to the buyer’s centroids and be excluded. This reflects the realistic scenario where buyers seek task-relevant data.
> Exploring new-class discovery in the low-similarity tails is future work, not part of this method.

---

### Meta-Review · Area_Chair_s9CR · 2026-01-11

**Summary:**

The paper proposes FSTS, a lightweight, training-free method for selecting core subsets from a target dataset using class centroids computed from a small source (buyer) dataset. While reviewers acknowledge the practical motivation in data-exchange scenarios and empirical gains at low selection ratios, major concerns include: (1) limited technical novelty—FSTS combines established ideas without significant algorithmic innovation; (2) experimental validation restricted to small, same-distribution splits (CIFAR-10, KITTI Road), which fails to reflect true transfer or large-scale settings; and (3) ambiguity around the “transfer learning” framing, as source and target come from the same dataset. Although the rebuttal clarifies the intended buyer–vendor scenario and justifies design choices (e.g., middle-band selection, median centroids), the contribution is widely viewed as incremental engineering rather than a scientific advance suitable for ICLR.

**Reviewer Concerns:**

Addressed by rebuttal:

Clarified that the setting is intra-task pre-acquisition filtering (not classical domain transfer), with source = buyer’s private data, target = vendor pool.
Justified use of same-dataset splits as a controlled simulation of real-world data-market conditions.
Provided theoretical and empirical support (citing Sorscher et al., NeurIPS 2022) for selecting the 40–60% similarity band to balance diversity and relevance.
Explained that median centroids and middle-band selection mitigate noise, redundancy, and OOD samples without extra computation.
Acknowledged computational constraints limiting model/dataset scale but emphasized algorithmic efficiency (no training, one-pass feature extraction).
Still outstanding:

Lack of novelty: All reviewers agree FSTS repackages known centroid-based and coreset ideas. Authors concede no architectural novelty, positioning the work as a benchmark—but this is seen as insufficient for ICLR.
Weak evaluation scope: Experiments remain on small datasets with synthetic distribution shifts; no cross-domain (e.g., DomainNet) or large-scale (e.g., ImageNet) validation.
Ambiguous contribution: The dataset curation and scenario framing are viewed by some (e.g., Reviewer Pm7n) as minor data processing rather than a meaningful scientific contribution.

**Reviewer Scores:**

Reviewer pmje (initial: 4 – marginally below threshold): Likely maintains 4. Found responses reasonable but still sees limited novelty and narrow evaluation.

Reviewer KBm9 (initial: 2 – reject): Likely raises to 3 or 4 after clarification that the setting isn’t classical transfer learning and equation/efficiency issues were addressed. May still question empirical scope.

Reviewer Pm7n (initial: 2 – reject): Unconvinced by rebuttal; explicitly stated authors “did not directly address” core concerns. Likely stays at 2 or moves only to 3 if persuaded by revised manuscript.

Reviewer Dt5h (initial: 4 – marginally below): Already sympathetic; likely moves to 5 (weak accept) given thorough rebuttal on failure cases, selection strategy, and scenario realism.

---

### Decision · Program_Chairs · 2026-01-26

Reject